# A mixed-methods comparison of gender differences in alcohol consumption and drinking characteristics among patients in Moshi, Tanzania

**Alena Pauley**[1], **Mia Buono**[1], **Kirstin West**[1], **Madeline Metcalf**[1], **Sharla Rent**[1,2], **Joseph Kilasara**[3,4], **Yvonne Sawe**[3], **Mariana Mikindo**[3], **Blandina T. Mmbaga**[3,4,5], **Judith Boshe**[3,4], **João Ricardo Nickenig Vissoci**[1,6], **Catherine A. Staton**[1,6]*

1 Duke Global Health Institute, Duke University, Durham, North Carolina, United States of America, 2 Duke Department of Pediatrics, Duke University Medical Center, Durham, North Carolina, United States of America, 3 Kilimanjaro Christian Medical Center, Moshi, Tanzania, 4 Kilimanjaro Christian Medical University College, Moshi, Tanzania, 5 Kilimanjaro Clinical Research Institute, Moshi, Tanzania, 6 Duke Department of Surgery, Duke University Medical Center, Durham, North Carolina, United States of America

* catherine.lynch@duke.edu

**Data Availability Statement:** Data are only available upon reasonable request, as participants did not consent to public data publishing, and data

## Abstract

Excessive alcohol use stands as a serious threat to individual and community well-being, having been linked to a wide array of physical, social, mental, and economic harms. Alcohol consumption differs by gender, a trend seen both globally and in Moshi, Tanzania, a region with especially high rates of intake and few resources for alcohol-related care. To develop effective gender-appropriate treatment interventions, differences in drinking behaviors between men and women must be better understood. Our study aims to identify and explore gender-based discrepancies in alcohol consumption among Kilimanjaro Christian Medical Center (KCMC) patients. A systematic random sampling of adult patients presenting to KCMC's Emergency Department (ED) or Reproductive Health Center (RHC) was conducted from October 2021 until May 2022. Patients answered demographic and alcohol use-related questions and completed brief surveys, including the Alcohol Use Disorder Identification Test (AUDIT). Through purposeful sampling, 19 individuals also participated in in-depth interviews (IDIs) that focused on identifying gender differences in alcohol use. Quantitative data was analyzed in RStudio through descriptive frequencies, proportions, ANOVA, and Chi-squared tests, while IDIs were analyzed in Nvivo following a grounded theory approach. During the 8-month data collection timeline, 676 patients were enrolled. Men and women patients at KCMC's ED and RHC were found to have significant differences in their alcohol use behaviors. For our quantitative data, this included lower average AUDIT scores among women (average [SD] AUDIT scores were 6.76 [8.16] among ED men, 3.07 [4.76] among ED women, and 1.86 [3.46] among RHC women). A subsequent IDI analysis revealed greater social restrictions around women's drinking and more secretive alcohol use behaviors for where and when women would drink. For men, excess drinking was normalized within Moshi, tied to men's social interactions with other men, and generally motivated by stress, social pressure, and despair over lack of opportunity. Significant gender

transfer requires a written agreement approved by Kilimanjaro Christian Medical Centre Ethics Committee and the National Institute for Medical Research (Tanzania). Data inquiries can be sent to Gwamaka W. Nselela at gwamakawilliam14@gmail.com.

**Funding:** This project was funded by the Duke Global Health Institute Graduate Student funds (AMP), and the Josiah Trent Foundation (21-06 to CAS). These two financial awards funded the salaries of JK, YS, and MMi as research assistants hired specifically for this study. No other authors received specific funding for this work. Infrastructure built by NIH grant (R01 AA027512 to CAS) was used to support the data collection process for this grant to understand gender-related aspects of alcohol use at KCMC. The funders had no role in study design, data collection and analysis, decision to publish, or preparation of the manuscript.

**Competing interests:** The authors have declared that no competing interests exist.

differences in drinking behaviors were found, primarily influenced by sociocultural norms. These dissimilarities in alcohol use suggest that future alcohol-related programs should incorporate gender in their conceptualization and implementation.

## Introduction

Alcohol use is a leading risk factor for death and disability-adjusted life years (DALYs) worldwide, accounting for over 3 million deaths each year [1,2]. Globally, alcohol use and alcohol-related harm are increasing, especially in low-and middle-income countries [3]. For example, the World Health Organization (WHO) Africa region consumes, on average, 20% more alcohol per day (40.0g/day) than the global average (32.8g/day), with Tanzania ingesting especially large quantities. The rate of heavy episodic drinking in Tanzania (7.7% among women and 33.4% among men 15 years or older) is almost twice that of neighboring countries [1]. Moshi, a popular tourist town located at the base of Mount Kilimanjaro in Northern Tanzania, has particularly high rates of alcohol use, which have also been increasing in recent years as compared to nearby regions [4–6]. This increase is influenced by a strong drinking culture and a custom of early alcohol initiation in minors for members of the Chagga ethnic group, who constitute the majority of local inhabitants [7,8]. Additional contributing factors are alcohol's ready availability mixed with its low cost and a recent increase in disposable income among local inhabitants [4,7,9].

Globally, alcohol consumption patterns and alcohol-related harms are dissimilar among men and women [10–15]. On average, men consume alcohol in higher quantities and more frequently than women, with a higher prevalence of Alcohol Use Disorder (AUD) worldwide [11,16,17]. Men are also more likely to engage in injurious behaviors like drunk driving, violence, and crime while under the influence [18,19]. While women consume less alcohol, drinking can reduce inhibitions and awareness of risk, increasing the likelihood of women being subject to unsafe scenarios like violence or sexual assault, which in turn increases the risk of acquiring sexually transmitted diseases [20–23]. For women who are pregnant, alcohol use also can introduce harm to themselves and their unborn child [24,25]. Sociocultural factors and norms are also known to impact alcohol consumption and behaviors. For example, among youth in both Tanzania and nearby Kenya, young people who spent time around those already consuming alcohol or encouraging alcohol use were found to have higher rates of intake themselves [7,26].

In Tanzania specifically, little is known about how or why alcohol use differs between genders. The data that does exist indicates that while intoxicated, men are more likely to exhibit violent behaviors, while women are more likely to be victims of violence. Men in Tanzania display a greater prevalence of alcohol use and abuse, a greater risk of injury after drinking, are more likely to incur road-traffic injuries, and have more public displays of alcohol intake than women [13,15,27]. This lessened and more private intake of alcohol from women may lie in religious, cultural, and social underpinnings that condemn female drinking in African cultures [28]. In contrast, alcohol consumption among women in the region increases the risk of experiencing sexual violence and contracting sexually transmitted infections [29]. Estimates for alcohol abuse have ranged from 7% among women with partners and 22.8% among men in the general population [4] to an AUD prevalence of 38.7% among men and 13.1% among women patients at a local outpatient clinic [30].

Alcohol-related care services are often not equitably utilized by gender, with women users disproportionately failing to receive care and appropriate treatment [31]. This trend is echoed

within Moshi in currently unpublished data from an ongoing clinical trial aimed at reducing alcohol misuse [32,33]. Designing and implementing alcohol-reduction interventions that incorporate gender differences in use can help minimize this inequity and more effectively care for women users [34]. As such, to reduce the burden associated with alcohol, more research about how and why alcohol intake varies between genders in Moshi is needed. Addressing this knowledge gap, this study aims to better understand key gender differences in alcohol use among patients in Moshi, Tanzania, by [1] quantitatively comparing measures of consumption and the relative risks of alcohol misuse among the patient populations studied, and [2] qualitatively exploring the distinctions between men and women's alcohol use behaviors.

## Methods

### Overview

This was a sequential explanatory mixed-methods study that combined quantitative survey score data and qualitative semi-structured in-depth interviews (IDIs). Quantitative data was collected prior to IDIs to guide qualitative data sampling, with all data collection procedures occurring from October 2021 until May 2022 at the Kilimanjaro Christian Medical Centre (KCMC) in Moshi, Tanzania (Fig 1). In total, 676 patients participated in survey questionnaires, and of those individuals, 19 were selected, via purposeful sampling, to participate in IDIs addressing alcohol use.

### Setting

This study was based within the Kilimanjaro Christian Medical Center (KCMC), a large referral and teaching hospital that serves over 1.9 million people [6]. KCMC is located in Moshi, an

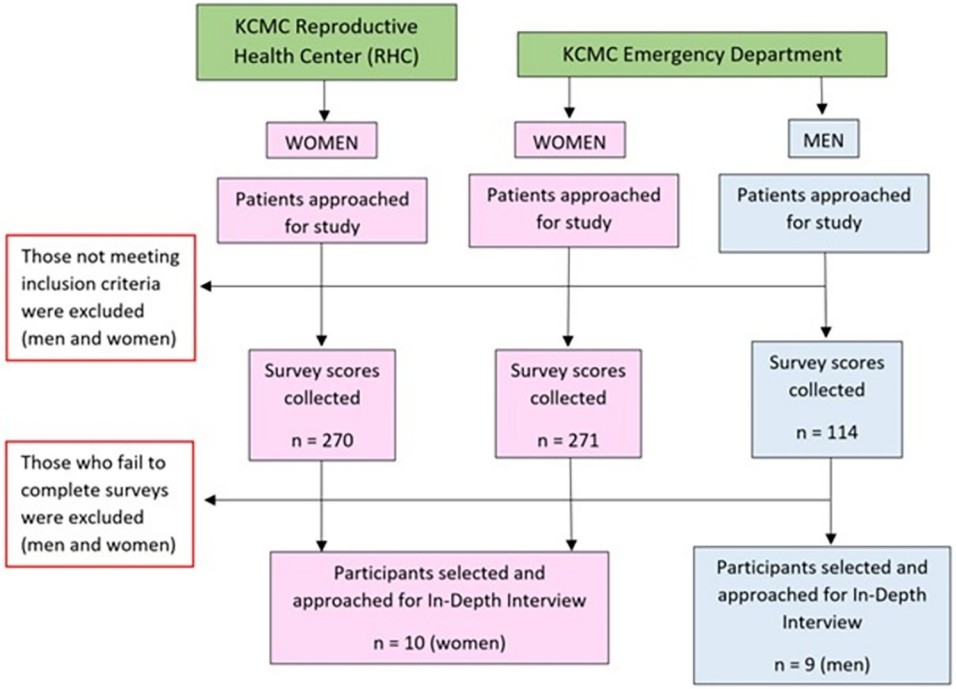

**Fig 1. Study design overview.**

urban town of over 200,000 residents in Northern Tanzania bordering Kenya and Kilimanjaro National Park. This study operated within two clinical units at KCMC, the Emergency Department (ED) and the Outpatient Unit for Gynecology, commonly referred to as the Reproductive Health Centre (RHC).

KCMC's ED serves as the referral unit for all injury patients in the Kilimanjaro region. The intoxicating effects of alcohol have long been implicated in the onset of serious injuries, such as road crashes or violent assault, which necessitate emergency care [35–37]. As a result, EDs globally tend to see a higher proportion of patients reporting alcohol misuse than is typically found in the general population [38,39]. As our long-term goal with research line is to reduce alcohol-related harm in this region, the high concentration of injury patients at KCMC's ED makes this clinical unit an ideal location as it allows a closer look at risky alcohol users and use behaviors.

KCMC's RHC serves as the referral unit for all women seeking gynecological care within the Kilimanjaro region. The RHC's large women patient population was chosen to facilitate a deeper understanding of women's drinking patterns while also providing a safe, gynocentric environment for collecting the sensitive information needed for this study. Data was collected in parallel at KCMC's ED and RHC throughout the 8-month enrollment timeline. Conducting this research at these locations allowed us to more accurately compare alcohol use behaviors, risky drinking in particular, among both men and women patients.

## Participants

All enrolled participants met the following eligibility criteria: 1) were 18 years of age or older, 2) had the capacity to give informed consent, 3) received initial care at KCMC's Reproductive Health Center or Emergency Department, 4) were conversant in KiSwahili, and 5) were not prisoners. Pregnancy was not an exclusion criterion for the study. Capacity to provide informed consent was defined as being medically stabilized, clinically sober, and well enough to complete the survey verbally on their own. For those who were extremely ill or injured upon initial presentation, the research team re-evaluated the patient within 24 hours of arriving at KCMC or before discharge, whichever came first. Those who remained unable to consent within this time frame were excluded from study participation.

At the time of enrollment, patients verified that they had not previously participated in this study either in the ED (male and female patients) or the RHC (female patients). This ensured each participant was only enrolled once. As this study was also conducted relatively early in the COVID-19 pandemic, patients who tested positive for COVID-19 were not approached for the safety of the data collection team.

## Procedures

All data were collected in the local language of KiSwahili by a team of three Tanzanian research assistants (two women and one man) who had been hired specifically to work on this study. The final group was chosen based on their expertise in research and strong interpersonal and leadership skills.

For data collection, the male research assistant surveyed and interviewed all men participants, and the two female research assistants surveyed and interviewed all women participants. This gender-matching between research assistants and interviewees was done to encourage open and honest reporting of patients' experiences with or opinions on alcohol based on local culture and research team experience [40]. Prior to data collection, thorough instruction on Good Clinical Practices and extensive study-specific training was provided to the Tanzanian team, including an overview of qualitative data collection methods. All members of the

Tanzanian research team were fluent in both English and KiSwahili and held a college degree or higher. The Tanzanian research team held a diverse set of competencies that promoted the collection of robust and reliable data, such as clinical nursing skills and numerous years of previous research experience specifically within the field of alcohol-related research.

## Quantitative data

**Sample size calculations.** During the initial study design, it was hypothesized that a final sample size of 587 patients would be required to determine if a significant difference in the prevalence of risky drinking, defined as a score of 8 or greater on the Alcohol Use Disorder Identification Test (AUDIT) scores exists between a) men and women KCMC Emergency Department (ED) patients and b) women patients at the KCMC ED and KCMC Reproductive Health Center (RHC) (Table 1). From the Emergency Department, 94 participants (47 men and 47 women) were surveyed to compare across genders the proportion of those who screened positively for risky drinking behavior with 80% power and 90% confidence. Likewise, 540 female participants (270 from the ED and 270 from the RHC) were surveyed to compare the prevalence of those who were positive for risky drinking behavior between these two units with 80% power and 90% confidence. The final sample size was 587, as survey score data collected from female ED patients were used in both analyses.

However, as of early December 2021, two months into the eight-month data collection timeline, the proportion of AUDIT scores $\geq 8$ among ED patients was approximately 40% for women and 45% for men, a difference of only 5% as compared to 15% difference upon which the original sample size calculations were based. This meant that to maintain 90% confidence and 80% power, 1,200 patients would need to be enrolled to determine the difference in proportions. While study funds and the data collection timeline limited this doubling in sample size, with IRB amendment approval, the study's targeted enrollment goals were increased based on the re-estimated prevalence. Thus, enrolling as many patients as was feasible within the original logistical study bounds resulted in a of 676 total participants by the conclusion of this study timeline.

**Procedures.** To collect as representative of a sample as possible in the ED and RHC, a systematic sampling method was employed. Patients seeking care at KCMC's EMD or RHC were enrolled Monday through Friday from 10:00 am until 6:00 pm local time, except for Tanzanian holidays. While enrollment of women patients was consistent throughout the entire data collection period, enrollment of men patients was paused from January 1st, 2022, until March 31st, 2022, pending expanded sample size regulatory approvals.

Within the RHC, which sees a large daily volume of women patients, every third patient listed on the general intake registry log was approached and offered study participation. Given the nature of RHC care, which often requires follow-up visits, a month into data collection, approximately a fifth of the daily RHC patient population had already been approached for study participation. As such, when sampling individuals from the intake registry, if a woman

**Table 1. Initial sample size calculations.**

| Location | Patient Gender | Expected Prevalence of AUDIT $\geq 8$ | Power | Confidence Level | Sample Size (comparison between men women in ED) | Sample Size (comparison between women in RHC & ED) | Total Enrollment |
|---|---|---|---|---|---|---|---|
| KCMC RHC | Women | 0.10 | 80% | 90% | N/A | 270 | 270 |
| KCMC ED | Women | 0.15 | 80% | 90% | 47 | 270 | 270* |
| | Men | 0.30 | 80% | 90% | 47 | N/A | 47 |
| | | | | Expected Participants | 94 | 540 | 587 |

had previously been asked about study participation, her name was skipped, and the research assistant then approached every third patient starting from the following individual.

At the ED, which sees significantly fewer women patients than men patients, every woman, but every 3rd man on the triage registry was approached. This was done to maintain planned enrollment goals and a representative, systematic random sampling of patients. Each of the three research assistants primarily enrolled one patient population (ED men, ED women, and RHC women). Halfway through data collection (once 135 women patients from both the ED and RHC had been collected), the two women research assistants switched clinical units to minimize any bias that may have arisen as a result of differences in their style of patient interaction or information extraction.

Patients were only approached once, and all were given the option to decline study involvement or terminate their participation early if they chose. All patients were approached in a quiet, private location only once medically stabilized. Here, an overview of the study, including the study goals, procedures, potential risks, and benefits, was explained. If, after this discussion, the patient was willing to participate, consent was obtained via their handwritten signature. For illiterate participants, consent was recorded through their written initials or a cross mark depending on their ability level. Surveys were administered orally by the same-gender research assistants so that patients of all literacy levels could participate, and responses were recorded into a secure Research Electronic Data Capture (REDCap) database. In rare instances where survey collection was interrupted, some surveys were left incomplete (n = 21). Incomplete surveys were included in the analyses but were not counted as part of the final sample needed for determining differences in prevalence.

**Instruments.** Quantitative surveys consisted of five main components: (1) basic demographic data, (2) self-reported alcohol use data consisting of scaled multiple choice or binary yes-no questions, (3) the Alcohol Use Disorder Identification Test (AUDIT), (4) the Drinkers' Inventory of Consequences (DrInC), and (5) the nine-item Patient Health Questionnaire (PHQ-9). AUDIT ranges from 0 to 40 and is a commonly used survey tool for measuring alcohol consumption and alcohol-related problems [41,42]. Both locally and globally, patients scoring greater than or equal to 8 are earmarked clinically significant for harmful or hazardous drinking (HHD) [42–46]. Patients with HHD represent individuals whose alcohol intake is detrimental to their physical well-being and require further alcohol-related clinical care and support [47]. As such, the prevalence of HHD (defined as AUDIT $\geq$ 8) was a primary cut-off point in this analysis. During study enrollment, participants who screened positive for HHD were referred for further care at KCMC. The AUDIT tool has previously been cross-culturally adapted, psychometrically validated, and clinically tested in the local context [46].

Alcohol use was also measured through self-reported alcohol consumption-related questions, asking participants how much and how often they consumed alcohol, what types of alcohol they preferred, and how much money they typically spent on alcohol per week. Participants were asked to report their alcohol consumption in terms of standard drinks, with one standard drink being defined as any beverage containing 10 g of pure alcohol, in line with WHO guidelines [48]. All non-survey tool questions were reviewed, revised, translated, and pilot tested by the Tanzanian research team prior to data collection. Of note, while this study focuses on gender differences, patients self-identified according to their biological sex. Given that there is little reported gender diversity in Tanzania, for the purposes of this analysis, those identifying as men were categorized as male, and those identifying as women were categorized as female.

**Analysis.** Gender differences in alcohol consumption and alcohol-related problems were assessed quantitatively through an exploratory analysis of AUDIT scores, the prevalence of HHD, and demographic and self-reported alcohol consumption data. All categorical data were

analyzed using descriptive frequencies and percentages, which included all variables except for AUDIT scores and age. AUDIT scores and age, being continuous variables, were analyzed as means with standard deviation. As alluded to above, AUDIT scores were dichotomized according to HHD status; scores of 8 or greater were classified as 'HHD,' while scores less than 8 were 'not HHD.' Except for age, income, and educational attainment measures, missing data were minimal for all variables analyzed. The age question was mistakenly omitted in the first several surveys and was added a week into data collection. Slightly more missing data are present in the questions related to income and educational attainment as participants were more hesitant to disclose this information to the research staff.

Unlike our qualitative data, quantitative data were compared across three groups: ED women, ED men, and RHC women rather than by gender alone. This was done to a) identify which clinical unit had the highest incidence of unhealthy alcohol users and b) provide more accurate descriptions of the two women patient populations as the RHC and ED women populations held significantly distinct demographic and alcohol use-related characteristics. Differences in the reported statistics among ED men, ED women, and RHC women were measured through Chi-squared tests or analysis of variance (ANOVA) as appropriate. All statistical analyses were conducted in RStudio (version 1.4) using user-created and validated R-Packages.

## Qualitative data

**Sample.**    Of all participants who completed the quantitative survey, a small subset was selected for participation in semi-structured IDIs. Twenty (or until saturation was reached) IDIs were initially anticipated to be collected, ten from ED men patients, five from ED women patients, and five from RHC women patients to facilitate gender-balanced perspectives on alcohol use. The men ED population reached saturation (defined as the absence of new themes and information following three consecutive interviews) in nine interviews. Thus, 19 IDIs in total were completed.

IDIs were used in this study given the highly sensitive and stigmatizing nature of the study topic, especially for women participants. The one-on-one interview structure encouraged participants to share their thoughts more freely while also helping to ensure their privacy and confidentiality. IDI procedures were first initiated during survey collection. At the time of the quantitative survey, if a research assistant identified an individual they thought would be an excellent candidate for an IDI, they asked if the subject would be willing to participate. Those invited were purposefully chosen to represent diverse demographic backgrounds (including age, marital status, education level and occupation, tribe, and religion), perspectives on, and personal experiences with alcohol. IDI participants were also selected to speak to trends related to risky drinking that arose from preliminary quantitative findings. One example is that women who were either divorced or widowed were associated with above-average alcohol intake early in the data collection period. Subsequently, a woman who had been recently divorced and had a high alcohol intake was purposefully asked to participate in an IDI. To ensure diverse representation and minimize any unintentional bias in sampling, the characteristics of IDI participants were reviewed monthly by the study lead, and any needed changes in the patient sampling were implemented at the following IDI selection.

If agreeing, the research assistant obtained the patients' phone numbers with their consent and scheduled a later time to meet. All interviews were held in private rooms within KCMC and were conducted by a same-gender interviewer who had an established relationship with the patient as they had previously spoken with the patient in-depth during survey collection. The goals of the research study were communicated again before interviews commenced, and a small fund of 5,000 TSH (~2 USD) was given to participants as a transportation

reimbursement. All interviews were audio-recorded and generally lasted between 60 to 100 minutes, with a break and snacks offered midway through.

**Instruments.** As with the surveys, all interview questions were created in English and translated into KiSwahili. These translations were then reviewed for appropriate phrasing and syntax, subsequently revised, and then pilot tested by the Tanzanian research team to ensure cultural appropriateness, relevancy, and retention of the original meaning. The interview guide consisted of open-ended questions with built-in probes. Additional probing questions were added on a case-by-case basis by the Tanzanian research team if a participant said something unclear, contradictory, or warranted further explanation. The guide was developed using a team-based approach. It was structured and organized across the six following domains: (1) effect on and expectations of the community, (2) men's use, (3) women's use, (4) gender differences in use, (5) use during pregnancy, and (6) recommendations for future interventions. The qualitative data in this manuscript pulls primarily from domains 2 through 4 to specifically explore how gender impacts alcohol consumption and use behaviors. Important themes regarding alcohol use and depression arose from the first seven interviews, so several questions related to these concepts were added at this point and were included in all following interviews. Except for this addition, the guide remained the same for all interviews.

**Analysis.** IDIs were analyzed using an applied thematic, grounded theory approach [49]. As female drinking behaviors in this region have received little prior research attention, a grounded theory approach was best suited for this study as it allowed new themes to arise and be sufficiently explored. A codebook, which was only accessible to the qualitative research team, was developed by the main data analyst based on the first four interviews following a mix of deductive and inductive coding schemes. The initial codebook was discussed with all members of the Tanzanian research team, and changes were made based on the received feedback, ensuring content validity and cultural accuracy. The codebook was used as a dynamic document and was updated as new themes emerged from the data. After revisions to the codebook were made, previous transcripts were revisited and recoded if necessary to encompass newly identified themes that emerged.

In partnership with the main analyst, the Tanzanian research team was trained on qualitative analysis and interview coding using Nvivo 12. The initial interviews were independently coded in four separate documents by the main analyst and the three members of the Tanzanian research team. These documents were then compared to establish an agreement on the coding strategy and codebook development. When disagreement arose between researchers, the research team discussed the codes in question until a consensus was reached. This process was repeated until 80% agreement was obtained among the four analysts, which occurred after three interviews were coded and reviewed [50]. After a high rate of internal consistency in coding was obtained amongst the four initial coders, the primary analyst used the final codebook to code the remaining 16 interviews. The final coding was approved by the analysis team. Content memos were created per each emerging theme and code, summarizing the findings in an ongoing fashion. The content memos served as a basis for discussion and feedback to the entire research team.

## Research ethics

Prior to data collection, ethical approval was obtained from the Duke University Institutional Review Board, the Kilimanjaro Christian Medical University College Ethical Review Board, and the Tanzanian National Institute of Medical Research. As much as possible, data was maintained in a de-identified manner and shared by data share agreement. Personal health information was used for screening and enrollment, but data were collected, stored, and analyzed in a de-identified manner.

## Results

### Quantitative

Between October 11th, 2021, and May 31st, 2022, all eligible patients present during study hours were approached, with 655 patients (Table 2) completing the surveys. Few individuals declined study participation; however, as reported by the Tanzanian research team, women were more likely than men to do so, with the primary reasons being that (a) they did not wish to discuss their alcohol use and (b) concern for their privacy.

Following the enrollment goals for each of the three patient populations, this study sample was composed primarily of women (82.6%). Most participants were also Christian (80%), employed (57%), and living with a partner either in a registered (50%) or unregistered (12%) marriage. RHC women had the greatest proportion of young patients, with 37% being between the ages of 25 and 34.

Across the three patient populations, the highest average [SD] AUDIT scores belonged to ED men (6.76 [8.16]), followed by ED women (3.07 [4.76]), and RHC women (1.86 [3.46]) (Table 3). ED men also had the highest prevalence of HHD (38%) across all patients, although a significant percentage of ED women (17%) still had AUDIT scores $\geq$ 8 (Table 3). RHC women had the lowest percentage of individuals with HHD (7.4%) across the three groups (Table 3).

In other markers of alcohol use, men continued to score above both women populations. ED men spent the most money on alcohol per week (4.4% of ED men, 0.7% of ED women, and 0.4% of RHC women spent between 50,001 to 1000,000 TZS per week), drank in the largest quantities (4.4% of ED men, 0.4% of ED women, and 0% of RHC women drank more than 6 standard drinks per sitting), and drank the most frequently (3.5% of ED men drank multiple times per day, but neither ED nor RHC women reported drinking more than daily). Interestingly, while men consumed the most, ED men and ED women answered affirmatively that they have attempted to quit drinking previously in roughly equal (51% for women and 52% for men) proportions, and men were also the most likely (89%) to believe that alcohol use was unhealthy (compared to 71% among ED women and 65% among RHC women).

While men had the highest rates of consumption, unhealthy alcohol users were present among both women patient populations. For example, 3.0% and 1.1%, of ED and RHC women, respectively, consumed 5 or more standard drinks in a sitting, and 3.7% of ED women and 1.1% of RHC women reported drinking alcohol every day.

**Qualitative.**   Of the 655 individuals who had complete surveys, 19 (RHC women, n = 5; ED women, n = 5; ED men, n = 9) participated in an IDI. Most IDI participants were Christian (78.9%), their ages ranged from 20 to 70 years, and they held a variety of education levels that stretched from primary education to a college degree. Almost half (9 out of 19) of the respondents were living with a partner in a registered marriage at the time of the interview, five were never married, two were divorced or separated, two were living with a partner in an unregistered marriage, and one was widowed. Only one participant was pregnant at the time of the interview, and five participants did not consume alcohol regularly. The intake of the other respondents ranged from 1 to 2 bottles 1 to 2 times per week to 3 to 4 bottles 5 to 6 times per week.

Alcohol use behaviors differed between men and women across all major themes (how, why, when, where, and what) assessed (Table 4). In general, respondents reported that men had greater control around their drinking, meaning greater ability and access to control their own drinking behaviors. They also drank more publicly and in greater quantities with higher alcohol-content drinks. For men, alcohol use was encouraged in social situations and was influenced by the potential for gaining social power or despair over a lack of life opportunities.

**Table 2. Study population demographics.**

| Demographics by Population Type | Overall, N = 655[1] | ED Women, N = 271[1] | RHC Women, N = 270[1] | ED Men, N = 114[1] | P Value |
|---|---|---|---|---|---|
| **Age Category** | | | | | <0.001 |
| 18 to 24 | 110 (18%) | 47 (19%) | 53 (21%) | 10 (9.5%) | |
| 25 to 34 | 159 (26%) | 41 (16%) | 92 (37%) | 26 (25%) | |
| 35 to 44 | 106 (17%) | 42 (17%) | 45 (18%) | 19 (18%) | |
| 44 to 54 | 94 (16%) | 47 (19%) | 32 (13%) | 15 (14%) | |
| Over 55 | 137 (23%) | 75 (30%) | 27 (11%) | 35 (33%) | |
| Missing/refused per block | 49 / 655 | 19 / 271 | 21 / 270 | 9 / 114 | |
| **Average Age** | 41.7 (18.7) | 46.1 (22.8) | 36.4 (12.7) | 45.2 (16.4) | 0.101 |
| **Personal Income Category (TZS per month)** | | | | | 0.001 |
| 0 to 50,000 | 205 (32%) | 110 (41%) | 68 (25%) | 27 (25%) | |
| 50,001 to 100,000 | 44 (6.8%) | 17 (6.3%) | 15 (5.6%) | 12 (11%) | |
| 100,001 to 150,000 | 56 (8.7%) | 17 (6.3%) | 29 (11%) | 10 (9.4%) | |
| 150,001 to 200,000 | 91 (14%) | 34 (13%) | 37 (14%) | 20 (19%) | |
| > 200,000 | 248 (39%) | 92 (34%) | 119 (44%) | 37 (35%) | |
| Missing/refused per block | 11 / 655 | 1 / 271 | 2 / 270 | 8 / 114 | |
| **Religion** | | | | | 0.187 |
| None | 11 (1.7%) | 5 (1.8%) | 3 (1.1%) | 3 (2.6%) | |
| Christian | 522 (80%) | 218 (80%) | 222 (82%) | 82 (72%) | |
| Muslim | 121 (18%) | 47 (17%) | 45 (17%) | 29 (25%) | |
| Other | 1 (0.2%) | 1 (0.4%) | 0 (0%) | 0 (0%) | |
| Missing/refused per block | 0 / 655 | 0 / 271 | 0 / 270 | 0 / 114 | |
| **Highest Educational Attainment** | | | | | <0.001 |
| None | 52 (8.7%) | 30 (12%) | 7 (2.9%) | 15 (13%) | |
| Primary | 182 (31%) | 82 (34%) | 61 (25%) | 39 (35%) | |
| Secondary | 133 (22%) | 48 (20%) | 64 (27%) | 21 (19%) | |
| College | 170 (29%) | 65 (27%) | 87 (36%) | 18 (16%) | |
| Graduate | 13 (2.2%) | 4 (1.7%) | 2 (0.8%) | 7 (6.2%) | |
| Vocational | 45 (7.6%) | 13 (5.4%) | 20 (8.3%) | 12 (11%) | |
| Missing/refused | 60 / 655 | 29 / 271 | 29 / 270 | 2 / 112 | |
| **Marital Status** | | | | | 0.013 |
| Never Married or Single | 135 (21%) | 55 (20%) | 56 (21%) | 24 (21%) | |
| Living with a partner, not in a registered marriage | 79 (12%) | 26 (9.6%) | 41 (15%) | 12 (11%) | |
| Living with a partner in a registered marriage | 327 (50%) | 128 (47%) | 140 (52%) | 59 (52%) | |
| Divorced or Separated | 3 (4.7%) | 16 (5.9%) | 6 (2.2%) | 9 (8.0%) | |
| Widowed | 81 (12%) | 46 (17%) | 27 (10%) | 8 (7.1%) | |
| Missing/refused | 2 / 655 | 0 / 271 | 0 / 270 | 2 / 114 | |
| **Employment Status** | | | | | <0.001 |
| Employed | 371 (57%) | 127 (47%) | 192 (71%) | 52 (46%) | |
| Unemployed | 215 (33%) | 111 (41%) | 50 (19%) | 54 (47%) | |
| Student | 69 (11%) | 33 (12%) | 28 (10%) | 8 (7.0%) | |
| Missing/refused | 0 / 655 | 0 / 271 | 0 / 270 | 0 / 114 | |
| **Tribe** | | | | | 0.070 |
| Chagga | 329 (50%) | 126 (46%) | 146 (54%) | 57 (50%) | |
| Iraq | 21 (3.2%) | 9 (3.3%) | 8 (3.0%) | 4 (3.5%) | |
| Maasai | 25 (3.8%) | 13 (4.8%) | 7 (2.6%) | 5 (4.4%) | |
| Mmeru | 29 (4.4%) | 14 (5.2%) | 11 (4.1%) | 4 (3.5%) | |

*(Continued)*

**Table 2.** (Continued)

| Demographics by Population Type | Overall, N = 655[1] | ED Women, N = 271[1] | RHC Women, N = 270[1] | ED Men, N = 114[1] | P Value |
|---|---|---|---|---|---|
| Muha or Non-African | 6 (1.0%) | 3 (1.1%) | 3 (1.1%) | 0 (0%) | |
| Nyaturu | 17 (2.6%) | 7 (2.6%) | 10 (3.7%) | 0 (0%) | |
| Pare | 70 (11%) | 27 (10.0%) | 25 (9.3%) | 18 (16%) | |
| Sambaa | 26 (4.0%) | 7 (2.6%) | 10 (3.7%) | 9 (7.9%) | |
| Sukuma | 34 (5.2%) | 10 (3.7%) | 16 (5.9%) | 8 (7.0%) | |
| Other African | 98 (15%) | 55 (20%) | 34 (13%) | 9 (7.9%) | |
| Missing/refused | 0 / 655 | 0 / 271 | 0 / 270 | 0 / 114 | |

[1]n / N (%); Mean (SD).

In comparison, IDI participants remarked that most women drank less, consumed lighter beers and wines, and were more restricted as to where and when it was acceptable for them to drink. This restriction was in part because of traditional gender roles keeping women more closely tied to the home and family life and also the concern for them to incur physical harm while drinking. For women, motives to drink were most closely tied to relationship stress and social pressure.

**How use differs by gender.** When asked how alcohol use differs between men and women, most IDI participants reported that men drank more than women, a trend that was facilitated by men having greater agency to drink, and men's drinking being more culturally normalized.

Overwhelmingly, respondents agreed that "*alcohol use among men is high compared to women*" (IDI #12, 34M, AUDIT = 14), but with the caveat that especially in recent years, this has been changing: "*sometimes women drink more alcohol than men. This is because women have [more] economical power than men nowadays, so are the ones that ruling the world.*" (IDI #5, 48M, AUDIT = 4)

Even still, for most, pre-existing community and familial norms appeared to limit women's alcohol consumption as compared to men. Traditional gender roles meant that men had greater freedom to drink alcohol, as explained in IDI #16.

"*Women are obliged to stay at home for a long time for the nurturing and taking care of children and family at large, so it's not easy for a woman to find time every day to go out with friends to drink alcohol. . .After [men] come out of job they don't have a lot to do at home like taking care of kids, they leave that all to women and mostly they instead go out with friends to drink. . .when a man comes from work he can pass by home and see the family leave some money and off he goes to have a drink outside. . .men have an ample time to relax, enjoy and that's when they get to drink alcohol.*"–IDI #16, 24F, AUDIT = 0

This sentiment was taken one step further by several male respondents who suggested that because of these responsibilities, some women are denied the option to drink at all; "*Woman are not even allowed to drink . . .because they get drunk easily and, once drunk, they will not be able to do home chores*" while "*men drink more than women because they are free and have money, so they can buy alcohol whenever they want*" (IDI #2, 35M, AUDIT = 19). As illustrated by these quotes, in comparison to men, women appeared to lack control over their ability to drink. This lack of agency is also confounded by having fewer funds, less free time, and more intensive home responsibilities.

**Table 3. Alcohol use characteristics.**

| Alcohol Use Characteristics by Population Type | Overall, N = 655[1] | ED Women, N = 271[1] | RHC Women, N = 270[1] | ED Men, N = 114[1] | P-Value |
|---|---|---|---|---|---|
| **Alcohol Preferences** | | | | | <0.001 |
| None | 258 (39%) | 100 (37%) | 143 (53%) | 15 (13%) | |
| Beer | 136 (21%) | 66 (24%) | 41 (15%) | 29 (25%) | |
| Changaa, Dadii, Gongo, or Piwa | 8 (1.2%) | 5 (1.8%) | 0 (0%) | 3 (2.6%) | |
| Light Beer | 60 (9.2%) | 20 (7.4%) | 21 (7.8%) | 19 (17%) | |
| Liquor/Spirits | 12 (1.8%) | 3 (1.1%) | 1 (0.4%) | 8 (7.0%) | |
| Mbege (banana-based beer) | 83 (13%) | 43 (16%) | 24 (8.9%) | 16 (14%) | |
| Ulanzi (bamboo-based liquor) | 5 (0.8%) | 2 (0.7%) | 1 (0.4%) | 2 (1.8%) | |
| Wine | 85 (13%) | 30 (11%) | 38 (14%) | 17 (15%) | |
| Missing/refused | 8 / 655 | 2 / 271 | 1 / 270 | 5 / 114 | |
| **Drinking Frequency** | | | | | <0.001 |
| 0 times/week | 269 (41%) | 101 (37%) | 146 (54%) | 22 (19%) | |
| 1–2 times/week | 289 (44%) | 130 (48%) | 100 (37%) | 59 (52%) | |
| 3–4 times/week | 66 (10%) | 28 (10%) | 17 (6.3%) | 21 (19%) | |
| 5–6 times/week | 6 (0.9%) | 2 (0.7%) | 2 (0.7%) | 2 (1.8%) | |
| Every day | 17 (2.6%) | 10 (3.7%) | 3 (1.1%) | 4 (3.5%) | |
| Multiple times a day | 4 (0.6%) | 0 (0%) | 0 (0%) | 4 (3.5%) | |
| Missing/refused | 4 / 655 | 0 / 271 | 2 / 270 | 2 / 114 | |
| Refused/Do not know | 2 / 653 (0.3%) | 0 / 271 (0%) | 1 / 269 (0.4%) | 1 / 113 (0.9%) | |
| **Drinks Consumed per Sitting** | | | | | <0.001 |
| 0 standard drinks | 268 (41%) | 102 (38%) | 145 (54%) | 21 (19%) | |
| 1–2 standard drinks | 264 (40%) | 119 (44%) | 87 (32%) | 58 (51%) | |
| 3–4 standard drinks | 97 (15%) | 42 (15%) | 34 (13%) | 21 (19%) | |
| 5–6 standard drinks | 16 (2.4%) | 7 (2.6%) | 3 (1.1%) | 6 (5.3%) | |
| >6 standard drinks | 6 (0.9%) | 1 (0.4%) | 0 (0%) | 5 (4.4%) | |
| Missing/refused | 4 / 655 | 0 / 271 | 1 / 270 | 3 / 114 | |
| **Weekly Alcohol Expenses (TZS)** | | | | | <0.001 |
| 0–10000 | 531 (81%) | 228 (84%) | 238 (88%) | 65 (58%) | |
| 10001–50000 | 110 (17%) | 41 (15%) | 29 (11%) | 40 (35%) | |
| 50001–100000 | 8 (1.2%) | 2 (0.7%) | 1 (0.4%) | 5 (4.4%) | |
| Missing/refused | 6 / 655 | 0 / 271 | 2 / 270 | 4 / 114 | |
| **Attempted Quitting** | | | | | <0.001 |
| No | 340 (52%) | 132 (49%) | 166 (61%) | 42 (37%) | |
| Yes | 298 (46%) | 137 (51%) | 102 (38%) | 59 (52%) | |
| Missing/refused | 17 / 655 | 2 / 271 | 2 / 270 | 13 / 114 | |
| **Reason for Quitting for the 298 patients who Attempted:** | | | | | 0.003 |
| Family | 17 / 298 (5.7%) | 9 / 137 (6.6%) | 3 / 102 (2.9%) | 5 / 59 (8.5%) | |
| Financial | 24 / 298 (8.1%) | 7 / 137 (5.1%) | 4 / 102 (3.9%) | 13 / 59 (22%) | |
| Health | 105 / 298 (35%) | 52 / 137 (38%) | 37 / 102 (36%) | 16 / 59 (27%) | |
| Personal | 128 / 298 (43%) | 60 / 137 (44%) | 49 / 102 (48%) | 19 / 59 (32%) | |
| Spiritual | 23 / 298 (7.7%) | 8 / 137 (5.8%) | 9 / 102 (8.8%) | 6 / 59 (10%) | |
| Other | 1 / 298 (0.3%) | 1 / 137 (0.7%) | 0 / 102 (0%) | 0 / 59 (0%) | |
| **Alcohol Use perceived as Unhealthy** | | | | | <0.001 |
| No | 179 (27%) | 74 (27%) | 92 (34%) | 13 (11%) | |
| Yes | 468 (72%) | 192 (71%) | 175 (65%) | 101 (89%) | |
| Missing/refused | 8 / 655 | 5 / 271 | 3 / 270 | 0 / 114 | |

*(Continued)*

**Table 3.** (Continued)

| Alcohol Use Characteristics by Population Type | Overall, N = 655[1] | ED Women, N = 271[1] | RHC Women, N = 270[1] | ED Men, N = 114[1] | P-Value |
|---|---|---|---|---|---|
| **Sought Treatment for Alcohol Use** | | | | | <0.001 |
| No | 606 / 652 (93%) | 251 / 269 (93%) | 260 / 269 (97%) | 95 / 114 (83%) | |
| Yes | 45 / 652 (6.9%) | 17 / 269 (6.3%) | 9 / 269 (3.3%) | 19 / 114 (17%) | |
| Missing/refused | 4 / 655 | 3 / 271 | 1 / 270 | 0 / 114 | |
| **AUDIT Score** | 3.22 (5.36) | 3.07 (4.76) | 1.86 (3.46) | 6.76 (8.16) | <0.001 |
| **HHD Status (AUDIT > = 8)** | 109 / 655 (17%) | 46 / 271 (17%) | 20 / 270 (7.4%) | 43 / 114 (38%) | <0.001 |

[1] n / N (%); Mean (SD).

In alignment with men's alcohol intake being higher, there were more examples of men who displayed dependent or problematic drinking behaviors–men who "*can't be okay without drinking*" (IDI #1, 24F, AUDIT = 0). Moreso, these examples were largely normalized–drinking "*too much. . . seems normal in the community*" (IDI #1, 24F, AUDIT = 0). One woman said:

"*[Men] drink alcohol as routine and it has now become compulsory for some of them to take a drink everyday of their life to complete their daily activities. For example, the men with heavy physical jobs but even doctors and businessmen also take alcohol every morning before going to work. I have witnessed a man who wouldn't go to his workplace before drinking because otherwise his hands would be shaking and he cannot touch anything and that's his routine. . .He was a doctor back in the village I was living.*"–IDI #3, 20F, AUDIT = 14

**Motivations to drink.** Among men and women, stress was the leading factor in alcohol initiation and progression into unhealthy drinking habits. This was followed by social pressure, and for men, social power. It is important to note that these factors are often intertwined within interviews, for example, social pressures could be seen as a form of relationship stress, and relationship stress is a form of social pressure.

Speaking first to stress, many respondents agreed that "*most people drink alcohol in the community due to stress. . .alcohol is no longer considered as a refreshing drink, rather than a stress*

**Table 4. IDI alcohol consumption themes and subthemes.**

| Themes | Sub-themes |
|---|---|
| How Consumption Differs | Men consume larger quantities of alcohol<br>Traditional gender roles impact alcohol consumption<br>Male drinking is more culturally normalized |
| Motivations to Drink | Relationship stress<br>Lack of opportunity<br>Social pressure<br>Social power |
| When Alcohol Intake Occurs | The best time to drink is after completing daily tasks<br>Women with families are too busy with household tasks to drink<br>Women with families are encouraged to drink after her kids are asleep |
| Where Alcohol Intake Occurs | Men drink in public locations to socialize with others<br>Public drinking for women creates a risk physical harm<br>Women drink privately to avoid social stigma |
| What Types of Alcohol is Consumed | Ability of Alcohol to Intoxicate the Drinker<br>Social clout<br>Wealth and socioeconomic status |

*remover substance*!" (IDI #5, 48M, AUDIT = 4). The causes of stress usually stemmed from conflicts in relationships–"*in town people drink too much because of stress related to love affairs, you find that there is conflict between them, especially cheating; this lead to excessive alcohol use to relieve the stress*" (IDI #4, 33M, AUDIT = 10)–or financial strain–"*life has become so difficult, and people cannot afford life expenses*" (IDI #5, 48M, AUDIT = 4).

For women especially, conflicts in relationships arose as a significant stressor and reason for alcohol initiation.

"*Where I live so many women live with stress because of being abandoned by men or men are there but they don't take care of the family. . .something which leads a lot of women to drinking too much alcohol every day. . .myself I have lived well with my husband but after a long illness he has run away from home and he has left me no money. If I didn't know God and hold onto him, I could end up drinking too much alcohol because of stress until I die*" (IDI #13, 41F, AUDIT = 0)

Another individual reported,

"*women drink alcohol to reduce stress when she has arguments with her husband, she decides to drink too much alcohol so that when she comes home, she will not talk to her husband anymore she will only fall to sleep.*" (IDI #6, 42F, AUDIT = 4)

The theme of relationship stress was present for males as well, but was not as strong of a factor as it was for females. IDI #7 (45F, AUDIT = 9) said "*when one has misunderstanding with his wife, he can get drunk to relieve the anger.*"

Limited career opportunities that led to financial stress and subsequent poor coping strategies, on the other hand, facilitated problematic drinking behaviors primarily among men. This disproportionate effect is in part because men are traditionally seen as the economic providers in Tanzanian culture.

"*Most young men drink too much alcohol. . .because they don't have any future about their life. This is associated with lack of employment, people finishing school but there is not a job to do which makes them frustrated.*" (IDI #8, 41M, AUDIT = 12)

A lack of professional opportunity seemed to impact young and older men alike; another participant, IDI #2 (35M, AUDIT = 19) noted that the previous year he was "*preoccupied with a lot of stress*" as his tourist company was "*not doing well*" and his "*mom was sick, and there was no money to take care of her.*" These stressors led him and other men to drink–what he described as "*a poor coping strategy*" (IDI #2, 35M, AUDIT = 19)–"*men drink more than women because they have stress and depression about life challenges*" (IDI #8; 41M, AUDIT = 12). One participant postulated that men relied on alcohol to relieve stress more than women did because

"*They don't want to speak about things which troubles them, they take out their problems through drinking alcohol. . .Men think when they talk about their problems it is a sign of weakness*" (IDI #15, 21F, AUDIT = 9)

Social pressure was another key factor driving alcohol use. For women, social pressures included peer pressure from social circles, engaging in drinking behaviors with their husbands, and using alcohol to obtain confidence in social situations. For example, one respondent said

"*it is [unusual] to find a women drink when the husband doesn't*" (IDI #12, 34M, AUDIT = 14). This was expounded upon by another participant, "*women drinks when their men drink as well, or sometimes they drink because of peer pressure or company. . .[they] drink because they want to relax, to gain confidence in social situations*" (IDI #11, 56M, AUDIT = 6). Further, many women interviewed expressed sentiments of alcohol allowing them to participate in activities "*they can't do while sober*" and "*to get rid of shyness*" (IDI #3, 20F, AUDIT = 14).

Beyond stress and social pressure, some men consumed alcohol because it was both a tool for connecting with other men and a symbol of social power, IDI #10 (54F, AUDIT = 4) for example, called alcohol use among men "*prestigious.*"

"*Men drink in order to please their friends or the people around them. If the surrounding people drink he too can drink to feel the sense of oneness with them and to be in one accord with the rest of the men around him.*" (IDI #3, 20F, AUDIT = 14)

IDI #15 (21F, AUDIT = 9) elaborates on this, saying "*there are some girls who prefer to be with a man who drinks alcohol. As I told you a man who drinks alcohol, he is working and can provide for the family. . .he is husband material.*"

Alcohol intake helped facilitate social connections for men in part because buying and consuming expensive alcohol was seen as a metric of financial power–

"*Men tend to drink much for showing off to his colleagues and gaining considerable prestige that he is wealthy*" as alcohol could cost "*up to three hundred thousand shilling per day*" (IDI #7, 45F, AUDIT = 9) (the equivalent of $130 USD).

Importantly, this was not the same for women; while women may have felt social pressure to drink, respondents remarked specifically that alcohol did not enable social interactions with other women in the same way that it did for men.

"*Men use alcohol as a social catalyst to help people meet and discuss issues. . .alcohol is not used among women as a source of them to meet and discuss. Is not as important as it is for men.*" (IDI #4, 33M, AUDIT = 10)

**When alcohol intake occurs.**   Overall, respondents reported that the times it was most acceptable to drink were after their daily tasks were completed, which in some cases differed between men and women. Because of the longer working hours of women with families, these individuals faced more restrictions as to when their drinking was seen as appropriate.

In general, respondents noted that men prefer to drink in the "*evening times to night hours after the work hours are over*" (IDI #3, 20F, AUDIT = 14), but for those who had heavier alcohol use or are "*addicted*" (IDI #17, 26M, AUDIT = 0) to alcohol, "*even in the morning you may find a man is already drunk*" (IDI #13, 41F, AUDIT = 0). Drinking in the early hours of the day was linked with being unemployed in IDIs for men, for example, "*for those who do not have permanent work they drink alcohol from morning they spend all day drinking alcohol*" (IDI #6, 42F, AUDIT = 4).

In contrast, most respondents agreed that women, especially those with families, faced more restrictions on when they should drink. This stemmed from their role as the primary family caretaker, and the reasons underlying this were two-fold: constraints on time, and avoiding drinking in front of children. Many noted that the full-time responsibility of taking care of the house and children left women little time to drink–"*for women, this is a challenge to get time free to drink because there are family duties waiting for her to handle*" (IDI #3, 20F,

AUDIT = 14). These restrictions occasionally were in some instances enforced by male respondents, for example, IDI #12 (34M, AUDIT = 14) remarked "*I told [my wife] that she should drink at the night when there are no activities at home.*"

Beyond the time restraint, women were also perceived as drinking alcohol when it was less visible by their family and community. Several remarked that mothers should not drink alcohol in front of their children–"*women drink at night when kids are asleep. . .kids are not supposed to see the mother drunk, it's a shame for a mother*" (IDI #4, 33M, AUDIT = 10)–also because "*they don't want kids to develop drinking habits*" (IDI #2, 35M, AUDIT = 19). One went so far as to say–"*[women] are scared to be seen drinking alcohol during the daylight so they use the darkness to their advantage*" (IDI #15, 21F, AUDIT = 9). Importantly, even for men who had children, these same restrictions were not mentioned by any IDI respondent regardless of gender.

These viewpoints on women's drinking were not held by all, however, one saying that the time when women drink "*don't differ much with men*" (IDI #18, 70M, AUDIT = 1) and another noting that compared to men "*time frame are just the same, from evening to night hours after they are done with work and business*" (IDI #16, 24F, AUDIT = 0).

**Where alcohol intake occurs.**   Even more than restrictions on appropriate times, strict rules for where a woman could drink arose among most IDIs. While men could drink anywhere but preferred bars and clubs, because of the potential for stigma and physical harm surrounding women's public drinking, women predominantly drank in the home due to their household duties.

There were few restrictions on where men could drink noted in IDIs–"*men are allowed to drink everywhere*" (IDI #12, 34M, AUDIT = 14), but the "*majority. . .like to go out to places like pubs, bars, and nightclubs*" (IDI #3, 20F, AUDIT = 14). Almost all participants reported that of all places for men to drink, "*most dislike to drink at their homes*" (IDI #7, 45F, AUDIT = 9), partially to form social connections but also as an escape from their home life–"*especially in marriages, when there's no peace at home it's easy for a man to drink too much. . .he spends so much time in bars to avoid staying at home*" (IDI #10, 54F, AUDIT = 4). One participant elaborated on this trend–

> "*Men prefer to drinking out of their home environment because the main reason is meetup with friends and discuss business ideas. Just imagine myself I'm always in the office, and when you go back home you meet a very same person that you always stay with, hahaha so it's better to drink in bars or hotel or another place that sells alcohol where you meet different people and exchange ideas.*" (IDI #4, 33M, AUDIT = 10).

Women respondents noted that bars and clubs were also desirable for men as a way to avoid interactions with their wife and meet other women–"*if someone has misunderstanding with his wife, he chooses to go bar and drink with other side women in order to get relief*" (IDI #7, 45F, AUDIT = 9).

For women though, the home was mentioned by the vast majority of respondents as the most appropriate place for women to drink–"*women drink at home, it's very rare to find them in bars or hotels drinking beers, morally is not allowed unless they go with her husband or boyfriend*" (IDI #4, 33M, AUDIT = 10). The remaining few either said that men and women drink in the same places as men ("*[women] now days drinks in the bars and pub because they go with their husbands. But in other settings women stays at home and drink alcohol*" (IDI #11, 56M, AUDIT = 6) or that women should not drink in the home "*because kids will be watching you*" (IDI #17, 26M, AUDIT = 0).

Respondents also said that women drank at home so that "*the community will not perceive them badly*" (IDI #6, 42F, AUDIT = 4), alluding to a greater stigma surrounding women's alcohol use.

> "*Finding a woman in a bar drinking alcohol in the early hours of the morning is not quite appropriate. A man can drink alcohol anywhere but a woman who drinks alcohol where many people can see her, they will look at her as a drunkard.*" (IDI #6, 42F, AUDIT = 4)

Rather, in stark contrast to men, the home was the best location for women's drinking because it is a private, "*secret*" setting where a woman can be with people "*she trusts,*" (IDI #3, 20F, AUDIT = 14) and likewise, where she can't be seen:

> "*Mostly women drink in hiding areas, [they] tend to hide themselves from either the husband or their relatives, they don't want others to know if they drink that much. A woman can leave like she is going to the shop to deceive her relative, while she is actually going to drink alcohol in the hiding where no one will notice her.*" (IDI #7, 45F, AUDIT = 9)

Another underlying factor as to why women did not drink at bars was the risk incurred on one's physical safety–"*liquor clubs it is not safe for a woman to drink alcohol because when they drink and get drunk, it often happens violence action against women*" (IDI #13, 41F, AUDIT = 0). Echoing this statement, a female college student reported:

> "*It is a risk for a woman to go to bars and clubs because of her safety, she may fail to get back home, get kidnapped, raped and sexually assaulted and lose her belongings... I have a female friend who drinks so much and that day she went to a club in town and on her way back she was robbed and beaten, and her belongings were taken. So, it is not a safe environment for women.*" (IDI #3, 20F, AUDIT = 14)

**What men and women drink and why.**　Men and women respondents both agreed that women typically prefer "*light alcohols with smaller percentages of alcohol*" whereas men liked "t*o drink strong liquors alcohol such as spirits*" (IDI #6, 42F, AUDIT = 4). These preferences were primarily influenced by alcohol's ability to intoxicate the drinker, social clout, and cost.

An overarching theme impacting men and women's choice of drink was that "*men drink for the aim of becoming drunk*"–IDI #12 (34M, AUDIT = 14), and since men have a higher alcohol tolerance, men preferred strong spirits so they could "*get drunk faster.*" One participant elaborated "*Most men drinks alcohol in Tanzania, whether they drink on public or secretly, but men drinks...Those who drinks strong beers the main reason is enjoyment and those who drinks local or least costive beers is because of stress or life hardship!*" (IDI #4, 33M, AUDIT = 10). In contrast, participants responded oppositely for women, noting that women "*don't want to get drunk*" (IDI #14, 47M, AUDIT = 27), so they widely gravitated towards lower-percentage and more mild alcohols. These choices for wines and "*light beers*" helped women to relax while also allowing them to complete their "*daily activities at home*" (IDI #14, 47M, AUDIT = 27; IDI #11, 56M, AUDIT = 6).

When asked more about why men and women prefer certain beverages, respondents explained the impact drinks had on interpersonal perceptions and social clout. For men, one factor contributing to the preference of stronger alcohols was to show their dominance and importance within the community–"*men drink strong alcohol because want to prove to women that they are superior and above everything (high self-esteem)*" (IDI #14, 47M, AUDIT = 27). Another echoed this statement, saying "*once he uses strong alcohol...people will view him as a*

*civilized and rich person instead for those who drinks mbege and other local beers*" (IDI #18, 70M, AUDIT = 1). These sentiments on certain alcohol's social prestige were not mentioned in relation to women.

For men too, echoing the motivations to drink, social clout was heavily tied with wealth and socioeconomic status. This in turn impacted the type of alcohol they had the ability to buy–"*people with high income drink alcohol of high standards and are expensive but those with low-income drink local brew*" (IDI #14, 47M, AUDIT = 27). Men living in more rural areas and of lower socioeconomic status for example were often marked as drinking the "*local brew*" that was widely available, affordable, and lacked testing for alcohol content levels before being sold. One male participant described:

> "*Most men drink beer but there are those who drink local brew, so it depends with the income of that particular person. But sometimes even those who drink beers can fall under local brew once they are bankrupt*! *In the villages the story is different; there they prefer strong alcohol, but locally made because it's cheap and people have no work to do, you may find them drinking from morning until late nights.*" (IDI #5, 48M, AUDIT = 4)

## Discussion

This study aimed to explore gender-based differences in alcohol use characteristics between patients at KCMC's ED and RHC. To the best of our knowledge, this study is the first of its kind to specifically explore alcohol consumption behaviors stratified by gender, both quantitatively and qualitatively, in Moshi, Tanzania. While current literature has investigated alcohol consumption among injury patients in Moshi and has demonstrated gender-specific differences in alcohol use [4,30] as well as the potential influence of gender stigma on risky alcohol behaviors [51–54], our study goes one step further to provide an in-depth mixed-methods analysis on how alcohol consumption and use behaviors differ by gender. Our results indicate that for men only, alcohol use can facilitate social interactions and act as a boon to their overall social status. In contrast, women face greater socially sanctioned restrictions surrounding their drinking and can have less agency than men in their personal alcohol use behaviors. Finally, in light of these social factors, rates of alcohol use among women patients at KCMC EMD were significantly higher than at the RHC, with male patients likewise exhibiting especially high use compared to the surrounding Moshi community. For populations in this region experiencing alcohol-related harm, understanding key sociocultural gender differences in alcohol use can more appropriately inform, target, and tailor future alcohol-related interventions.

We found that for men, alcohol was viewed as a tool for connecting with other men and as a symbolic representation of their social power. This connotation encouraged men to drink, to do so in social settings like bars, and to consume expensive liquors as a marker of their economic status. Two recent studies in Moshi and one in neighboring Uganda [55] have similarly described alcohol as a form of "social currency" and "an important part of social celebrations" [56], however, without delineating whether this association is skewed by gender. This lack of delineation may stem from the aim of these studies, which focus on overall drinking culture as opposed to gender differences in drinking. Even still, a finding in Osaki's work that young men feel greater pressure to initiate alcohol intake from their peers than young women [53] parallels ours. This disproportionate peer pressure on men likely stems from alcohol use is more interlinked to men's social lives than women.

While specific data on male drinking and social power in the African context is lacking, this association may find its roots in traditional African cultural practices, where alcohol has been traditionally reserved for high-class men [57]. Among the Nyakyusa people of Southern

Tanzania [58], for example, men who were older and more socially respected were the primary consumers of beer. The reliance on alcohol as a social tool and symbol among men in Moshi likely contributes to higher continued rates of alcohol use. Concerningly, the fact that alcohol is interwoven into the fabric of male social life may likely make it even more difficult for problematic male drinkers to reduce or abstain from intake in this region.

In stark contrast to alcohol's interconnectedness with male social life, we found that women experience significant social restrictions around drinking and tend to have less agency than men in their alcohol use behaviors. Previous studies in Tanzania and neighboring regions have found social influences [12,13] affect alcohol use, but still there is a significant dearth of data on sociocultural-based gender roles and responsibilities impact women's alcohol use. Our findings did have strong parallels to what relevant research was available, specifically two studies based in Tanzania. Work by Griffin et al. found that women who drink experience disproportionate social stigma [52], and literature by Meier et al. found that drinking publicly is considered culturally inappropriate for women but not for men. In addition to social stigma and sanctions, this finding by Meier et al. is likely a result of the potential physical harm women risk of incurring while drinking in public spaces, as the link between alcohol use and gender-based violence has been well-established [59,60]. Interestingly, our finding stood opposite to Dumbili et al.'s work in Nigeria, where young women consumed excessive quantities of alcohol to gain social capital [12]. This may be because, while on the same continent, Nigeria and Tanzania have distinct cultural differences, meaning that social capital is likewise achieved differently.

An important consequence of the heightened restrictions around women's drinking is that it leads to secretive alcohol use behaviors, a finding that has a unique clinical implication for women's healthcare. As we found in this analysis, some women did not want others to see them consuming alcohol, which led them to drink at night, in private locations, or only around certain people. This alludes to a broader stigma around women's alcohol use, which is concerning as stigma has been associated globally [54,61–63] and within Africa [64] as a barrier to effective healthcare service delivery. Because women who drink in Tanzania have been shown to face greater stigma than men, this obstacle in alcohol use-related treatment delivery may be especially prominent for them [52]. Thus, women's secretive alcohol consumption, a behavior born out of stigma and social restrictions around their drinking, will likely make it difficult for healthcare workers to identify, diagnose, and treat women with unhealthy alcohol use.

In addition to the sociocultural and economic factors that serve as determinants for alcohol consumption, we found that alcohol misuse was highest within the ED at KCMC. While potentially harmful alcohol consumption was the most pronounced among male ED patients specifically (38%), female ED patients also exhibited higher rates of concerning alcohol use compared to their RHC counterparts (17% ED and 7.4% RHC). Comparing our findings with local estimates, Mitsunga and Larsen found that 7% of women with partners, 9.3% of women without partners, and 22.8% of men tested positive for alcohol abuse (alcohol abuse here defined as a CAGE score of 2–4) [4]. Mushi et al. observed the prevalence of AUDIT $\geq 8$ (HHD) to be 23.9% for outpatient primary healthcare patients in Moshi. When stratifying for gender, 38.7% of males and 13.1% of females tested positive for HHD [30] compared to our findings of HHD among 38% of ED men patients and 17% of ED women. Rates of HHD were comparable among men in these two studies, but for women, a significant difference could be seen. This suggests that KCMC's ED has an exceptionally high concentration of women with high-risk alcohol use compared to other populations in Moshi.

Our finding of higher rates of alcohol use among men versus women echoes current global trends [13,65,66]. Global literature also supports the increased rates of alcohol use and high-risk drinking behaviors within EDs compared to other hospital departments or wards. This is

because of alcohol's close association with trauma and injuries, potentially requiring more immediate medical care [36,67]. For those with limited access to care because of financial constraints, EDs may serve as one of the only options to obtain health services for minor alcohol-related consequences. Within the East Africa and LMIC context, previous research has also found elevated rates of alcohol use among ED patients in Tanzania, most often presenting with injuries resulting from road traffic incidents [68]. The high rates of harmful alcohol consumption among both men and women ED patients in Tanzania suggest that the ED may be an optimal site for alcohol-related interventions.

While alcohol-related harm was not central to this analysis, it is important to emphasize not only the negative impact excessive alcohol use can cause on the Moshi community but also the opportunities for reducing this burden moving forwards. The findings discussed in this manuscript–the high proportion of unhealthy alcohol users in KCMC's ED, the restrictions imposed on women's alcohol use and resulting secretive drinking behaviors, and the association between social life and alcohol use for men–all provide valuable insight that can be used to help shape future alcohol-reduction interventions more effectively.

First, our finding that the ED had a significantly high proportion of unhealthy alcohol users, especially women users, suggests that this clinical unit may be a good location to base alcohol-related interventions. This is especially important when considering our finding of women's limited agency and secretive behaviors around their alcohol use. The lack of agency women have around drinking may impede the delivery of alcohol-related treatments. However, an ED-based intervention could help efficiently target this high-risk group as part of their usual care. Thus, an ED-targeted intervention may benefit high-risk women within the community who may face greater social restrictions in accessing alcohol-related health services. Second, the significant dissimilarities in men's and women's alcohol use behaviors–from where, when, what, and why they drink–point to the need for creating alcohol-related interventions that are differentiated by gender. For example, as shown in our analysis, the reasons for drinking varied by gender. Men's reasons were more closely tied to social pressure and lack of opportunity, whereas for women, their reason for drinking was primarily stemmed from relationship stress. In reducing alcohol intake, the underlying causes of drinking must be addressed, and as this comparison shows, different sociocultural and environmental factors are important for men versus women in their alcohol use. Alcohol-reduction-related interventions and programs that are conscientious of gender differences in alcohol use behaviors may be more able to bring about a lasting change in consumption patterns.

## Strengths and limitations

To our knowledge, this was the first study based in Moshi dedicated to exploring the intersection between gender and alcohol use. The significant differences found between men and women's use behaviors imply that gender should be more closely considered in future alcohol-related treatments and policy planning. In considering this analysis, however, several limitations must be taken into account. Despite extensive measures at ensuring confidentiality, patient recall of alcohol use characteristics may have been implicated by recall bias, whereby patients could not accurately recount drinking behaviors, or were cautious of exposing their drinking habits to investigators. Moreover, women were more likely to decline study participation, with primary reasons being (a) they did not wish to discuss their alcohol use and (b) concern for privacy. Consequently, it is important to consider the possibility that our study did not capture the full extent of alcohol consumption among women, given the implications of gender stigma present in this context. That is, it may be that women are more reluctant to share the extensiveness of their drinking behaviors in fear of being stigmatized by community

and family members. This may have influenced the accuracy and validity of these results. Additionally, survey and IDI data were obtained from three different clinical settings, warranting replication and external validation. Although missing data was not significant, we must be cautious of the external validity of our results. We recommend that further studies should be conducted and replicated at additional clinics, with limited missing data and increased assurance of confidentiality and comfortability. Finally, the study from which this data draws lacks quantitative data on the social implications of alcohol use, which limits our ability to triangulate some of our qualitative findings. Future studies should thus aim to address this gap to ensure the validity of these results.

## Conclusion

Men and women patients at KCMC's ED and RHC were found to have significant differences in their alcohol use behaviors, including lower rates of consumption among women, greater social restrictions around women's drinking, and more secretive alcohol use behaviors, including where and when women could drink. Conversely, for men, excess drinking was normalized within the Moshi community, tied to men's social interactions with other men, and generally motivated by stress, social pressure, and despair over lack of opportunity. These dissimilarities indicate a need for future alcohol-related programs to incorporate gender in their design and implementation. Our findings can be used to make these programs more effective at reducing alcohol-related harm in the Kilimanjaro region.

## Supporting information

**S1 Checklist. STROBE statement—checklist of items that should be included in reports of observational studies.**
(DOCX)

**S2 Checklist.**
(DOCX)

**S1 File.**
(PDF)

**S2 File.**
(PDF)

**S3 File.**
(PDF)

**S1 Questionnaire.**
(PDF)

**S2 Questionnaire.**
(PDF)

## Author Contributions

**Conceptualization:** Alena Pauley, Blandina T. Mmbaga, Judith Boshe, Catherine A. Staton.

**Data curation:** Alena Pauley, Joseph Kilasara, Yvonne Sawe, Mariana Mikindo.

**Formal analysis:** Blandina T. Mmbaga, Judith Boshe, Catherine A. Staton.

**Funding acquisition:** Alena Pauley, Catherine A. Staton.

**Investigation:** Blandina T. Mmbaga, Judith Boshe, Catherine A. Staton.

**Methodology:** Alena Pauley, João Ricardo Nickenig Vissoci, Catherine A. Staton.

**Supervision:** Blandina T. Mmbaga, João Ricardo Nickenig Vissoci, Catherine A. Staton.

**Writing – original draft:** Alena Pauley, Mia Buono, Kirstin West, Madeline Metcalf.

**Writing – review & editing:** Alena Pauley, Mia Buono, Kirstin West, Madeline Metcalf, Sharla Rent, Catherine A. Staton.

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
