## [Decision Letter · Decision Letter 0]

6 Jul 2023

PGPH-D-23-00826

A Mixed-Methods Comparison of Gender Differences in Alcohol Consumption and Drinking Characteristics among Patients in Moshi, Tanzania

Dear Dr. Staton,

Thank you for submitting your manuscript to PLOS Global Public Health. After careful consideration, we feel that it has merit but does not fully meet PLOS Global Public Health’s publication criteria as it currently stands. Therefore, we invite you to submit a revised version of the manuscript that addresses the points raised during the review process.

We look forward to receiving your revised manuscript.

Kind regards,

Priya Ranganathan

Academic Editor

Journal Requirements:

3. Please provide separate figure files in .tif or .eps format only and remove any figures embedded in your manuscript file. Please also ensure all files are under our size limit of 10MB.

Additional Editor Comments (if provided):

This is an important and robust study with interesting findings. However, the manuscript needs major revisions. Please see the detailed comments below:

Please use the use the COREQ checklist, or other relevant checklists listed by the Equator Network, such as the SRQR, to ensure complete reporting of your study

ABSTRACT

Methods:

Add a sentence on how both sets of data were analysed

Results

'.....alcohol use behaviors including lower rates of consumption among women...' AUDIT scores do not indicate 'rates' of consumption

Please separate the quantitative and qualitative results. At the moment you report both in a single sentence and it will confuse the reader

MAIN PAPER

Please check manuscript for English and grammar. Examples of errors include 'While women intake less alcohol.....'; 'The data that does exist has indicated that while intoxicated, men are more likely to exhibit violent behaviors while women are more likely to be victims of violence.' ‘Standing also as contributing factors is alcohol’s ready availability….’

INTRODUCTION

The 'aim' is very broad. Please supplement with specific objectives

Line 98 - This paragraph is a repetition of the previous paragraph. Please discuss past literature that possibly explains the variation in alcohol consumption by gender, looking at studies within the study region or the country or even beyond.

Line 105 - This paragraph is the justification for this study and it needs to be expanded explicitly.

Line 133 - What does this phrase mean? “Injuries have long been associated with excessive alcohol use”.

METHODS

Is there a potential for selection bias- those with injuries are more likely to be consuming alcohol heavily? Also, did you exclude pregnant women attending the gynaecology clinic? If not, would that also be sample with potential selection bias i.e. pregnant women might be more likely to reduce/stop drinking?

'No women participants presented to both the ED and RHC'. How would you know if they presented at these two settings at two different time points during the course of recruitment?

'As surveys were collected at a single time point, there were no patients lost to follow-up' This sentence is superfluous.

'Quantitative surveys consisted of five main components: (1) basic demographic data, (2) self-reported alcohol use data......' Was #2 just a binary yes-no question or a set of questions? If it was the later, then please provide details.

What do you mean by the 'depression module' of PHQ-9. The whole PHQ-9 is a tool for identifying potential depression.

If you are not reporting PHQ-9 and DrInC in this manuscript then please do not describe in methods section. Suggest reporting atleast mean PHQ-9 and DrInC scores for each one of the groups.

Please move the following to the section related to data and not in the analyses: 'Consumption was also measured through self-reported alcohol consumption questions that asked participants how much and how often they consumed alcohol, what types of alcohol they preferred, and how much money they typically spent on alcohol per week.'

Line #279 There are no sample size calculations in qualitative studies. Please label this section as 'sample' and merge the content from 'procedures' below as it describes how you tried to achieve maximum variability in the sample on various parameters.

Ethics: How was consent recorded in illiterate participants? Was any care provided to those who screened positive on the AUDIT?

RESULTS

Tables 2 and 3: Since the denominator remains the same for each block, please do not report it for each cell in that block. Just report the missing value at the bottom of each block.

Table 1: Please report mean age (SD) as well. Why do you need to report both personal and household income? Please report just one i.e. the one which is most representative of SE status in this setting.

Please arrange the categories for all the variables in Tables 1 and 2 in a logical sequence e.g. None -> Primary  Secondary etc

Table 3: How have you merged drinking frequency (times/week) with intensity (multiple times a day)? For example the person drinking 1-2 times per week could be drinking multiple times on those days

Table 3: Quantity over what period/drinking session? Also, are all the drinks consumed as bottles? I would assume not. Especially the 'spirits'

Please also report mean (SD) and range for expenditure on alcohol.

Please arrange the response items logically (No followed by yes). Please ensure that you check all the variables and ensure that there is a logical reporting of the categories in each variable. Why is 'refused/do not know' not excluded as missing data for all these variables? Not doing so is leading to an inaccurate synthesis of data

Figures 2 and 3 are not adding anything to the narrative and can be excluded

IDI number is not required. For each quote please report, gender, age, and mean AUDIT score.

Line #743 - use appropriate terms. You cannot measure incidence through a survey.

What does "agency around drinking" mean?

DISCUSSION

The discussion is focused around the qualitative findings and some about the prevalence in the survey. What about all the other details related to prevalence of drinking etc.? None of that has either been discussed or triangulated with the qualitative findings

Reviewers' comments:

Reviewer's Responses to Questions

**Comments to the Author**

1. Does this manuscript meet PLOS Global Public Health’s publication criteria? Is the manuscript technically sound, and do the data support the conclusions? The manuscript must describe methodologically and ethically rigorous research with conclusions that are appropriately drawn based on the data presented.

Reviewer #1: Yes

Reviewer #2: Yes

2. Has the statistical analysis been performed appropriately and rigorously?

Reviewer #1: Yes

Reviewer #2: No

3. Have the authors made all data underlying the findings in their manuscript fully available (please refer to the Data Availability Statement at the start of the manuscript PDF file)?

Reviewer #1: No

Reviewer #2: Yes

4. Is the manuscript presented in an intelligible fashion and written in standard English?

Reviewer #1: Yes

Reviewer #2: No

5. Review Comments to the Author

Reviewer #1: The manuscript examines the prevalence and patterns of alcohol use and alcohol use disorders in a select population from Tanzania. It is a mixed methods study with a cross sectional survey and nested qualitative study. While the study is robust there are several gaps in reporting which will need to be addressed before the paper could be considered fro publishing.

ABSTRACT

Methods:

Add a sentence on how both sets of data were analysed

Results

'.....alcohol use behaviors including lower rates of consumption among women...' AUDIT scores do not indicate 'rates' of consumption

Please separate the quant and qual results. AT the moment you report both in a single sentence and it will confuse the reader

MAIN PAPER

Please check manuscript for English and grammar. Examples of errors include 'While women intake less alcohol.....'; 'The data that does exist has indicated that while intoxicated, men are more likely to exhibit violent behaviors while women are more likely to be victims of violence.'

INTRODUCTION

The 'aim' is very broad. Please supplement with specific objectives

METHODS

Is there a potential for selection bias- those with injuries are more likely to be consuming alcohol heavily? Alos, did you exclude pregnant women attentding the gynaecology clinic? If not, would that also be sample with potential selection bias i.e. pregnant women might be more likely to reduce/stop drinking

'No women participants presented to both the ED and RHC'. How would you know if they presented at these two settings at two different time points during the course of recruitment?

'As surveys were collected at a single time point, there were no patients lost to follow-up' This sentence is superfluous

'Quantitative surveys consisted of five main components: (1) basic demographic data, (2) self-reported alcohol use data......' Was #2 just a binary yes-no question or a set of questions? If it was the later, then please provide details

What do you mean by the 'depression module' of PHQ-9. The whole PHQ-9 is a tool for identifying potential depression

If you are not reporting PHQ-9 and DrInC in this manuscript then please do not describe in methods section. Suggest reporting atleast mean PHQ-9 and DrInC scores for each one of the groups

Please move the following to the section related to data and not in the analyses: 'Consumption was also measured through self-reported alcohol consumption questions that asked participants how much and how often they consumed alcohol, what types of alcohol they preferred, and how much money they typically spent on alcohol per week.'

Line #279 There are no sample size calculations in qual studies. Please label this section as 'sample' and merge the content from 'procedures' below as it describes how you tried to achieve maximum variability in the sample on various parameters

Ethics:How was consent recorded in illiterate participants? Was any care provided to those who screened positive on the AUDIT?

Tables 2 and 3: Since the denominator remains the same for each block, please do not report it for each cell in that block. Just report the missing value at the bottom of each block.

Table 1: Please report mean age (SD) as well. Why do you need to report both personal and household income? Please report just one i.e. the one which is most representative of SE status in this setting.

Please arrange the categories for all the variables in Tables 1 and 2 in a logical sequence e.g. None -> Primary  Secondary etc

Table 3: I am not sure how you have merged drinking frequency (times/week) with intensity (multiple times a day). For example the person drinking 1-2 times per week could be drinking multiple times on those days

Table 3: Quantity over what period/drinking session? Also, are all the drinks consumed as bottles? I would assume not. Especially the 'spirits'

Please also report mean (SD) and range for expenditure on alcohol

Please arrange logically (No followed by yes). Please ensure that you check all the variable and ensure that there is a logical reporting of the categories in each variable. WHy is 'refused/do not know' not excluded as missing data for all these variables? Not doing so is leading to an inaccurate synthesis of data

I do not think figures 2 and 3 are adding anything to the narrative and can be excluded

IDI number is not required. For each quote please report, gender,age, and mean AUDIT score

Line #743 use appropriate terms. You cannot measure incidence through a survey

The discussion is focussed around the qual findings and some about the prevalence in the survey. What about all the other details related to prevalence of drinking etc. None of that has either been discussed or triangulated with the qual findings

Reviewer #2: Line 98 This paragraph is a repetition of the previous paragraph. I was expecting the researcher discuss past literature that possibly explained the variation in alcohol consumption by gender, looking at studies within the study region or the country or even beyond.

Line 105 This paragraph is the justification for this study and it needs to be expanded explicitly.

Line 133 What does this phrase means? “Injuries have long been associated with excessive alcohol use”.

Line 154 What does this phrase means? “No women participants presented to both the ED and RHC”.

The Quantative result section was not coordinately written. I would suggest the researcher give an initial report from the ED for males and females, then go on to report the findings from the RHC.

Also, Figure 2 was not explained in any part of the results. Also what are the levels of significance of all the findings across this study? Report your p values.

Line 413 “In general, respondents reported that men had greater agency around their drinking, meaning greater ability and access to control their own drinking behaviors”. There is no such statement as having agency around drinking, use appropriate and comprehensive terms.

In the Qualitative result, if the participants were chosen purposely (Line 293-295) then the result for these categories variation in gender alcohol consumption and characteristics should be reported to reveal the gender differences across various socio-cultural and socio-economic population.

Line 688 “boon” or “boom”?

Report the Implication of this study and the limitations in a separate sub-headings from discussion.

6. PLOS authors have the option to publish the peer review history of their article (what does this mean?). If published, this will include your full peer review and any attached files.

**Do you want your identity to be public for this peer review?** For information about this choice, including consent withdrawal, please see our Privacy Policy.

Reviewer #1: **Yes: **Abhijit Nadkarni

Reviewer #2: **Yes: **Adeyemo Queen Esther

---

## [Editor Report · Decision Letter 1]

8 Sep 2023

A Mixed-Methods Comparison of Gender Differences in Alcohol Consumption and Drinking Characteristics among Patients in Moshi, Tanzania

PGPH-D-23-00826R1

Dear Dr Staton

We are pleased to inform you that your manuscript 'A Mixed-Methods Comparison of Gender Differences in Alcohol Consumption and Drinking Characteristics among Patients in Moshi, Tanzania' has been provisionally accepted for publication in PLOS Global Public Health.

Best regards,

Priya Ranganathan

Academic Editor
